# Impact of COVID-19 pandemic on health service utilisation and household economy of pregnant and postpartum women: a cross-sectional study from rural Sri Lanka

Sajan Praveena Gunarathna ,[1] Nuwan Darshana Wickramasinghe,[1] Thilini Chanchala Agampodi,[1] Indika Ruwan Prasanna,[2] Suneth Buddhika Agampodi[1]

¹Department of Community Medicine, Faculty of Medicine and Allied Sciences, Rajarata University of Sri Lanka, Saliyapura, Anuradhapura, Sri Lanka
²Department of Economics, Faculty of Social Sciences and Humanities, Rajarata University of Sri Lanka, Mihintale, Anuradhapura, Sri Lanka

**Correspondence to**
Dr Sajan Praveena Gunarathna;
sajaanpraveen7@gmail.com

## ABSTRACT

**Objectives** This study aims to describe how household economies and health service utilisation of pregnant and postpartum women were affected during the pandemic.

**Design** A cross-sectional study.

**Setting** This study was conducted in the Anuradhapura district, Sri Lanka.

**Participants** The study participants were 1460 pregnant and postpartum women recruited for the Rajarata Pregnancy Cohort during the initial stage of the COVID-19 pandemic.

**Primary and secondary outcome measures** Household economic (income, poverty, nutritional and health expenditures) and health service utilisation details during the COVID-19 pandemic were gathered through telephone interviews. Sociodemographic and economic data were obtained from the cohort baseline and analysed with descriptive and non-parametric analysis.

**Results** Out of the 1460 women in the sample, 55.3% (n=807) were pregnant and 44.7% (n=653) were postpartum women. Of the total sample, 1172 (80.3%) women participated in the economic component. The monthly household income (median (IQR)=212.39 (159.29–265.49)) reduced (median (IQR)=159.29 (106.20–212.39)) in 50.5% (n=592) families during the pandemic (Z=−8.555, p<0.001). Only 10.3% (n=61) of affected families had received financial assistance from the government, which was only 46.4% of the affected income. The nutritional expenditure of pregnant women was reduced (Z=−2.023, p=0.043) by 6.7%. During the pandemic, 103 (8.8%) families with pregnant or postpartum women were pushed into poverty, and families who were pushed into poverty did not receive any financial assistance. The majority of women (n=1096, 83.3%) were satisfied with the free public health services provided by the public health midwife during the pandemic.

**Conclusion** During the early stages of the pandemic, healthcare utilisation of pregnant women was minimally affected. Even before the country's current economic crisis, the household economies of pregnant women in rural Sri Lanka were severely affected, pushing families into poverty due to the pandemic. The impact of COVID-19

## STRENGTHS AND LIMITATIONS OF THE STUDY

⇒ We conducted telephone interviews during the COVID-19 pandemic with a large sample of pregnant and postpartum women who were participants in a maternal cohort in Sri Lanka.

⇒ We combined three large data sets (baseline and follow-up economic data and telephone interview data of the Rajarata Pregnancy Cohort study) to identify the changes in the household economy and health service utilisation with the pandemic.

⇒ The data quality and validity may be compromised due to the selection bias and the mode of interviews (telephone interviews), even though remote data collection is the alternative option available during a pandemic.

⇒ Although we identified that the sociodemographic characteristics are not statistically significant different between pregnant and postpartum women in the present sample, the two groups may be qualitatively different due to non-response during the pandemic.

and the aftermath on pregnant women will have many consequences if the policies and strategies are not revised to address this issue.

## INTRODUCTION

Pregnant and postpartum women in low-income and middle-income countries are among the most vulnerable groups considering the indirect impact estimates due to health system collapse and food insecurity during the COVID-19 pandemic.[1 2] Disrupted healthcare service delivery, accessibility and poor healthcare utilisation during the COVID-19 pandemic and the household economic crisis that followed in the aftermath are impediments to achieving Sustainable Development Goals (SDG) targets.[3 4] The pandemic was an unexpected catastrophe in

the global and national sociocultural and economic landscape affecting the daily well-being of households directly or indirectly.[5] [6] Many countries implemented stringent pandemic control measures, including national or local lockdowns,[7] which led household economies into a crisis and affected health service utilisation.[8]

Many countries recommended spacing clinic visits during pregnancy and following up remotely by phone; only reserving emergency care was provided at the hospitals.[9] Both high-income and low-income countries reported changes/reductions in maternal healthcare services during the COVID-19 pandemic. According to the available evidence, maternity care and birth practices changed in the USA, increased the number of home births, conducted prenatal visits through online platforms, and partners and support persons were excluded from birthing rooms leaving mothers unsupported.[10] Nepal reported a reduction of institutional childbirth by more than half, with rises in neonatal mortality and poor quality of care.[7] There has been a reduction in health service utilisation in North-East Ethiopia due to fear of pregnant women attending health institutions.[5] In addition, poor access to healthcare care and reduced perceived social support for pregnant women were reported around the globe.[11] Thus, the COVID-19 pandemic has adversely affected the utilisation of maternal healthcare services.

Few studies have cited the adverse impact of the COVID-19 pandemic on the household economy.[6] [8] These studies investigated the adverse effects on household economies in Nigeria,[12] changes in consumption patterns of Malaysian households,[13] losing jobs and declining household income in some Asia-Pacific countries—Cambodia, the Lao People's Democratic Republic, Malaysia, Indonesia, the Philippines, Myanmar, Thailand and Vietnam.[14] Economic burden during COVID-19 could be more problematic for households of pregnant women, since pregnancy expenditure was already a burden in many low-income and middle-income countries.[15] However, studies explicitly focusing on the COVID-19 impact on the household economy of pregnant women are lacking in the global evidence base.

Sri Lanka is a lower-middle-income country[16] where maternal health services are well developed and accessed by pregnant women via both the public (to a more significant extent) and the private health system.[17] [18] The government's free public health system delivers maternal care through a well-established network of primary healthcare officers, reporting 85.3% of pregnant women registered in recent years.[19] Also, the country is among the leads of lower-middle-income countries regarding maternal and child health SDGs. Sri Lanka has reached the fourth phase of the obstetric transition,[20] and the healthcare system of Sri Lanka is an excellent example of universal public health system coverage, disease prevention through evidence-based strategies, and using culturally appropriate manners for health promotion, etc.[18] The solid network of public health personnel, especially the public health midwives (PHMs) that involve in

domiciliary and clinic care in the field, has been an asset to the system.

The first local index case emerged on 11 March 2020, and lockdowns/curfews were initiated after 17 March 2020.[21] During the COVID-19 pandemic's initial stage, a low mortality rate due to COVID-19 was recorded with the high public health system standards in Sri Lanka.[21] The interim guidelines on maternal healthcare services were provided to sustain the services during the pandemic by the Family Health Bureau, Ministry of Health, the national focal point of maternal healthcare in the Ministry of Health, Sri Lanka.[22] [23] It provided expert guidance on the procedures for organising services, home visits, clinic services, communications and protective measures.[22] However, the COVID-19 pandemic caused a higher prevalence of anxiety and depression among pregnant women, requiring additional support from the family and field health officers.[24] This could be worse if the household economy were affected, since the income status was associated with the pregnancy expenditure[25] [26] and pregnancy nutritional status.[27] Available evidence indicates that the COVID-19 pandemic created an economic meltdown at the household level and the Sri Lankan economy.[28]

How many Sri Lankan maternal health services have been affected by the poor economic conditions of the household during the COVID-19 pandemic is not well explored. The empirical studies explicitly focusing on the impact of the COVID-19 pandemic on maternal healthcare service utilisation and the household economy of pregnant women in Sri Lanka are lacking. Assessing the effects of the COVID-19 epidemic on the household economy and health service utilisation of pregnant and postpartum women would yield valuable evidence for decision-making during pandemic situations. Thus, this study aimed to find whether the COVID-19 pandemic affected the household economy and health service utilisation of pregnant and postpartum women in rural Sri Lanka.

## METHODS
### Study design and study setting
We conducted a cross-sectional study via telephone interviews. The participants were the women in the economic component of the largest pregnancy cohort in Sri Lanka, the Rajarata Pregnancy Cohort (RaPCo).[29] [30] Anuradhapura district is the largest district in Sri Lanka (7179 km$^2$), with a current population of about 900 000, and most of the population (92.7%) belongs to the district's rural population. The agriculture industry is the primary source of income, and the median monthly household income is US$285.91.[31] Total health expenditure comprises 58% of government sources,[32] and maternal, postpartum and child healthcare is provided free of charge by the government sector through 22 Medical Officer of Health (MOH) areas and 275 PHM areas. Approximately 17000 pregnant women register at the PHM annually, and the

number of live births is about 15 000.[29] Further details on the study setting have already been published.[29]

## Study population and sample

The study population included pregnant women registered in the Anuradhapura district's maternal care programme, where the antenatal care coverage is 82.3%.[29] All pregnant women fulfilling the inclusion criteria were invited for the baseline assessment of the RaPCo study from 1 July 2019 to 30 September 2019 to participate in this study. The eligibility criteria of the RaPCo were as follows:

► Pregnant women registered at the PHM and visiting antenatal field clinics in the Anuradhapura district.
► Pregnant women whose permanent residence is in the Anuradhapura district and had not planned to leave the district until childbirth.
► Period of amenorrhoea/gestational age is less than 13 weeks by the time of recruitment.

Pregnant women with uncertain due dates and those who planned to leave the study setting for delivery were excluded from the study. The pregnant women for the study were recruited with the help of PHMs in each MOH area by organising a special clinic with the permission of the district's public health authorities. The original cohort included >90% of the study population. All pregnant women recruited were invited for the baseline economic study.

## Data collection

With the permission of pregnant women during recruitment of the RaPCo, we collected their telephone numbers to contact them to share the investigation results and follow-up data collection. We held a telephone interview using a pretested interviewer guide with close-ended questions and collected details on the routine utilisation of antenatal/postnatal health services and household financial information during the pandemic.[33] Data collection was based on five major categories: (1) whether income was affected during the outbreak, (2) financial assistance received, (3) the status of using maternal health services, (4) the assistance of the PHM and (5) the assistance of household members/neighbours (online supplemental file 1).

Five female medical graduates trained in telephone interview techniques, emphasising politeness, telephone etiquette, basic counselling skills and essential perinatal health information, carried out the interviews. If the first attempt failed, several attempts were made to contact pregnant women on the same day and subsequent days for the next 2 weeks. The second round of calls was done to reach those unavailable or busy the first time. Telephone interviews were conducted mainly during the pandemic and completed within 3 months (from April to June 2020). The duration of the interview was 4–10 min.

We explained the study objective and confirmed whether the study participation was voluntary. We obtained verbal consent during the interview apart from the written consent sorted at the recruitment in the RaPCo study. During the telephone interviews, we provided psychological support to pregnant and postpartum women where needed.

## Definition of measures

We presented the sample characteristics and third trimester per visit pregnancy expenditure between women delivered during (after 17 March 2020) and before (before 17 March 2020) the COVID-19 pandemic. We considered 17 March 2020 as the cut-off date to create this variable, since the national lockdowns/curfews began on 17 March 2020.[21]

To identify the impact of the COVID-19 pandemic on the household economy, we used monthly household income, poverty status and financial assistance during the pandemic as the main variables. Monthly household income was the total monthly earnings of all household members during the pandemic. The national (US$3.65) and extreme (US$2.15) poverty lines were used for poverty analysis. For the status of receiving financial assistance, we considered only the government financial assistance (value of US$26.55). It was the only financial aid provided by the government, provided monthly for 3 months and only for selected families living under the national poverty line.

Furthermore, we created a few variables, that is,
► Income reduction of $i^{th}$ household=income before COVID-19 pandemic of the $i^{th}$ household − income during COVID-19 pandemic of the $i^{th}$ household.
► Percentage of affected income over household income of the $i^{th}$ household=(reduced income of the $i^{th}$ household×100%)/household income of the $i^{th}$ household.

The same formula was used to create the percentage of received financial aid over affected income, the percentage of received financial aid over household income and the percentage of received financial aid over household expenditure.

Testing the differences in per visit pregnancy expenditure during the third trimester was made based on the similarity of sociodemographic profiles within two groups (online supplemental table 2), and all pregnant women who delivered during COVID-19 were in their third trimester. Moreover, the per month pregnancy-related nutritional expenditure was the amount spent for pregnant women's dietary intake per month.

To identify the status of health service utilisation during the pandemic, we used the following variables: missed clinics, number of missing clinic visits, whether changed the mode and place of childbirth and home visits of PHM, etc. Also, to measure the status of support received from the PHM and the family members, we used the Likert scale, where 1 denoted 'strongly dissatisfied' and 5 denoted 'strongly satisfied'.

## Data analysis

We entered data during the interviews using Microsoft Office Excel software and data management using the

same. Next, we combined the baseline data collected from the RaPCo and the economic study with this database and analysed data using SPSS V.27. Before the analysis, all monetary values were converted to the average exchange rate (US$ to Sri Lankan Rupees (LKR)) of the data collection period from April 2020 to June 2020 (ie, US$1=LKR 188.33).[34] The average exchange rate for the above period was based on the fact that there were no outliers, and the SD was only US$2.99.

Sample characteristics (including sociodemographic, economic and health-related information), health service utilisation information and household financial information during the pandemic were presented using univariate descriptive statistical measures. Since the data were not normally distributed (p<0.05) due to the Shapiro-Wilk test, we used non-parametric statistical tests for bivariate analysis, including the Wilcoxon Signed Ranks Mann-Whitney U test and $X^2$ goodness-of-fit test.

## Patient and public involvement

The telephone follow-up interviews for the RaPCo study were developed based on community engagement. Pregnant women recruited in RaPCo called the 24-hour helpline during the COVID-19 lockdown, expressing concerns about their health status, psychological issues and accessing help. The research team took note of these issues raised by the participants to conceptualise the telephone-based follow-up study focusing on financial, social and health problems and offered psychological support as well as appropriate interventions. The questions for the calls were also based on inquiries from the participants. While the participants were not directly involved in the design, their feedback and engagement influenced the development of the follow-up method.

**Table 1** Sociodemographic and economic characteristics of the study sample

| Characteristic | | Statistics (n=1460, 100%) |
|---|---|---|
| Age of the pregnant women (mean (SD)) in years | | 28.2 (5.5) |
| Ethnicity (n (%)) | Sinhalese | 1250 (85.6) |
| | Other minority ethnic groups* | 210 (14.4) |
| Religion (n (%)) | Buddhist | 1237 (84.7) |
| | Other religions† | 223 (15.3) |
| Educational status (n (%)) | Primary education | 15 (1.0) |
| | Junior secondary education | 63 (4.3) |
| | Senior secondary education | 1162 (79.6) |
| | Higher education | 220 (15.1) |
| Daily wage group (n (%)) | Yes | 296 (20.3) |
| | No | 1164 (79.7) |
| Income-generating members in the household | Only one person | 1180 (80.8) |
| | More than one person | 280 (19.2) |
| Income status in relation to the poverty line (n (%))‡ | Households below the national poverty line | 173 (11.8) |
| | Households below the extreme poverty line | 112 (7.8) |
| Per visit pregnancy expenditure during the first trimester (US$) (median (IQR)) | | 3.72 (0.88–7.88) |
| Per visit pregnancy expenditure during the second trimester (US$) (median (IQR)) | | 3.45 (0.88–8.50) |

*Tamil, Moor, Malays and others.
†Catholic/Christian, Hindu, Islam and others.
‡Income status before COVID-19 pandemic.

## RESULTS

We invited all pregnant women of the RaPCo (n=3374), and 1460 women participated in this study (response rate=43.3%). This was the only number contactable during the pandemic, and no woman opted out during the interview. The household income (p=0.743), household expenditure (p=0.512), religion (p=0.363), ethnicity (p=0.515), education level (p=0.361) and status of sexual and reproductive health education (p=0. 344) were not significantly different between the present study sample participants and other RaPCo participants.

## Sample characteristics

The mean (SD) age of the pregnant and postpartum women was 28.2 (5.5) years. Among the total sample (n=1460), 44.7% (n=653) of women delivered before the COVID-19 pandemic and 55.3% (n=807) during the COVID-19 pandemic. Table 1 shows the sociodemographic and economic characteristics of the study sample.

## Impact of COVID-19 pandemic on household economy

The pregnant and postpartum women's monthly household income showed a statistically significant difference (p<0.001) during the pandemic and the pre-COVID-19 period. The household income was affected in 50.5% (n=592) of families, and the percentage of reduced income in terms of household income was 74.6%. Though the monthly household income was reduced, there was no statistically significant difference (Z=−0.827, p=0.408) in household expenditure during and before the COVID-19 pandemic for the families of women who delivered during COVID-19 (n=807, 55.3%).

The majority (n=1111, 94.8%) had not received government financial assistance, and the obtained amount covered only 46.4% of the affected income (table 2).

Daily wage earners' (n=296, 20.3%) income was reduced by 80.0%. Only 14.5% (n=43) of families in the daily wage group received government financial assistance (US$26.55), covering 57.6% of affected income, 19.1% of household income and 33.5% of household expenditure.

**Table 2** COVID-19 impact on the household economy, pregnancy expenditure and status of receiving financial assistance

| Description | | | Statistics (n=1172, 100%) |
|---|---|---|---|
| Impact on household income | Families affected by income reduction (n (%)) | Yes | 592 (50.5) |
| | | No | 580 (49.5) |
| | Reduced amount (US$) | n (%) | 592 (50.5) |
| | | Median (IQR) | 106.20 (53.10–159.29) |
| | Percentage of reduced income over household income (%) | | 74.6 |
| Monthly household income (US$) changes | n (%) | | 592 (50.5) |
| | Income before COVID-19 (median (IQR)) | | 212.39 (159.29–265.49) |
| | Income during COVID-19 (median (IQR)) | | 159.29 (106.20–212.39) |
| | Statistically significant difference* | | P<0.001 |
| Households pushed into poverty (n (%)) | National poverty line | | 56 (4.8) |
| | Extreme poverty line | | 47 (4.0) |
| Financial assistance provided by the government (US$26.55) | Received financial aid among families pushed into poverty (n (%)) | | 2 (1.9) |
| | Received financial assistance among affected families (n (%)) | | 61 (10.3) |
| | Percentage of received financial aid over affected income (%) | | 46.4 |
| | Percentage of received financial aid over household income (%) | | 17.9 |
| | Percentage of household expenditure covered by the received financial aid (%) | | 29.9 |
| Per month pregnancy-related nutritional expenditure (n=807, 55.3%) | Before the COVID-19 period (median (IQR)) | | US$54.43 (37.17–79.65) |
| | During the COVID-19 period (median (IQR)) | | US$53.10 (37.17–79.65) |
| | Statistically significant difference* | | P=0.043 |
| Per visit out-of-pocket expenditure in the third trimester | Delivered before COVID-19 pandemic† (n=524, 44.7%) (median (IQR)) | | 4.82 (0.73–10.19) |
| | Delivered during COVID-19 pandemic‡ (n=648, 55.4%) (median (IQR)) | | 4.73 (1.02–9.64) |
| | Statistically significant difference* | | P=0.771 |

*Wilcoxon signed ranks test.
†The cost indicated here was incurred before the COVID-19 pandemic.
‡The cost indicated here was incurred during the COVID-19 pandemic.

Among the total sample, a considerable number of families (n=103, 8.8%) were pushed into poverty and extreme poverty due to the pandemic. The total affected amount (median (IQR)=US$106.20 (53.10–159.29)) was equivalent to 72.7% of their monthly household income. However, only 10.3% (n=61) of affected families had received financial assistance from the government, which was 54.3% of the affected income, 31.0% of the monthly household income and 40.4% of the monthly household expenditure. Financial assistance was not given to families with pregnant women who were pushed into poverty; in total, only 1.9% (n=2) of families who were pushed into poverty received financial assistance. The breakdown analysis of the impact of the household economy between pregnant (delivered during COVID-19) and postpartum women (delivered before COVID-19) was reported in online supplemental table 2.

A statistically significant difference (Z=−2.020, p=0.043) was noted in per-month pregnancy-related nutritional expenditure during and before the COVID-19 pandemic. The reduction in nutritional spending was 6.7% compared with early pregnancy. However, there was not a statistically significant difference (U=293.000, p=0.582) of pregnancy-related nutritional expenditure during COVID-19 outbreak between families which faced income loss and their counterparts.

There was no statistically significant difference (U=196680.00, p=0.771) in spending per visit during COVID-19 compared with pre-COVID-19. An extended analysis of pregnancy expenditure breakdown (direct medical (cost for consultation, medicine, laboratory investigation) and non-medical pregnancy expenditure (cost for travelling, food and refreshments, accompanying person)) was reported in online supplemental table 3.

### Impact of COVID-19 on routine health service utilisation of pregnant and postpartum women

Of the total sample, 14.2% (n=104) of pregnant women and 6.9% (n=40) of postpartum women reported missing clinics due to the COVID-19 pandemic. The median (IQR) number of missed clinics was 1.[1 2] The majority (n=1096, 83.3%) of the pregnant and postpartum women were satisfied with the service provided by the PHM

**Table 3** Impact of COVID-19 on health service utilisation of pregnant and postpartum women and support received by PHM and family

| Description | | | Statistics (n=1315, 100%) |
|---|---|---|---|
| Clinic visits* | Whether missed clinics (n (%)) | Yes | 144 (11.0) |
| | | No | 1171 (89.0) |
| | Number of missed clinics (median (IQR)) | | 1 (1-2) |
| Field health services from PHM | Whether PHM visited or did phone check-ups to see the pregnant women/infant (n (%)) | Yes | 1170 (89.0) |
| | | No | 145 (11.0) |
| | Number of visits per month (median (IQR)) | | 2 (1–3) |
| | Service provided by the PHM (n (%)) | Strongly satisfied | 7 (0.5) |
| | | Satisfied | 1096 (83.3) |
| | | Average | 188 (14.3) |
| | | Not satisfied | 18 (1.4) |
| | | Strongly not satisfied | 6 (0.5) |
| Childbirth | Whether changed the mode of childbirth (n (%)) | Yes | 4 (0.5)† |
| | | No | 728 (99.5) |
| | Whether changing the place of childbirth (n (%)) | Yes | 14 (1.9)‡ |
| | | No | 718 (98.1) |
| Support from husband/family and neighbours received during the pandemic (n (%)) | Very satisfied | | 16 (1.2) |
| | Satisfied | | 1092 (83.1) |
| | Average | | 161 (12.2) |
| | Dissatisfied | | 27 (2.1) |
| | Very dissatisfied | | 19 (1.4) |

*Indicated clinic visits are antenatal clinic visits for the women delivered during COVID-19 and postnatal/child health clinics for women delivered before COVID-19.
†Number of pregnant women who had to change the mode of childbirth
‡Number of pregnant women who had to change the place of childbirth.
PHM, public health midwive.

during the pandemic. Most women (n=1170, 89.0%) visited/received PHM phone check-ups, and the median (IQR) visits/phone check-ups were 2.[1–3] Furthermore, most pregnant (n=1092, 83.1%) and postpartum women were satisfied with the support/assistance received by their spouse/family members and neighbours during the COVID-19 pandemic. Only 0.5% (n=4) of pregnant women had to change the childbirth mode from normal vaginal delivery to lower segmant C-section, and 1.9% (n=14) women had to change the place of childbirth due to the travel restrictions of the curfew period (table 3).

There is no association between the health service utilisation of pregnant women and the income changes during the COVID-19 pandemic ($X^2$(df=1)=0.361, p=0.548). The breakdown analysis of the impact of health service utilisation between pregnant (delivered during COVID-19) and postpartum women (delivered before COVID-19) was reported in online supplemental table 4.

## DISCUSSION

This study assessed the COVID-19 pandemic impact on health service utilisation and the household economy of pregnant and postpartum women in rural Sri Lanka.

Since the study was based on the follow-up of a maternal cohort, we could compare the impact on the household economy with the pre-COVID-19 values. We found that the household income is reduced by 74.6% in most families. Totally, 103 families (8.8%) were pushed below the poverty line, while 19.5% of families already remained in the poorest group. We identified that the most vulnerable group was daily income earners, and the financial aid received was two-fifth, a small share compared with the income loss.

No changes were observed in per visit pregnancy expenditure between pregnant women who delivered during and before the COVID-19 period. However, minor differences were noted in the cost of medicine, the cost for accompanying persons and direct medical expenditures. We also detected a reduction in the nutritional expenditure of pregnant women during the COVID-19 period. Health services were minimally affected, and health service utilisation was not associated with income reduction. Further, pregnant and postpartum women were highly satisfied with the services provided by PHM.

 Gunarathna SP, *et al. BMJ Open* 2023;**13**:e070214. doi:10.1136/bmjopen-2022-070214

## Impact on household economy and pregnancy expenditure

Even though it is the early stage of the COVID-19 pandemic, most households' income was affected by the local and national pandemics. Almost half of the families experienced an income reduction; the reduced income was equivalent to three-quarters of the monthly household income. The significant number of families pushed into poverty (n=56, 4.8%) and extreme poverty (n=47, 4.0%) show the pandemic's severity. The most vulnerable group was daily income earners, who were one-fifth of the total sample. The reason was unavailability and the inability to seek work due to pandemics and curfews.[35] Daily income earners must spend 70.0% of per day's income per visit to antenatal care in rural Sri Lanka.[26] As reported in Pakistan,[36] this could impact pregnant women's mental health. Household income reduction and poverty due to the pandemic were common phenomena, as documented by many researchers.[37–39]

Food insecurity and changes in food consumption patterns due to income reduction during the COVID-19 pandemic were recorded in Kenya and Uganda,[40] Indonesia,[41] Tajikistan,[42] Thailand[43] and in multicountry surveys.[44] Although nutritional status is considered necessary in pregnancy, nutritional expenditure is reduced compared with early pregnancy due to income reduction. This could be a sensitive matter, since 16.5% of pregnant women were recorded as underweight and 14.5% as anaemic in the sample, as reported in other studies conducted in the same study setting.[45–47] Household food diversity and nutritious food intake are compromised at the early stage of food insecurity in the same population of pregnant women even before the pandemic.[48] Hence, necessary policies and mechanisms must be implemented to safeguard the food security of pregnant women at-risk during similar situations.

The burden of out-of-pocket expenditure has been highlighted despite accessing free services from government health facilities in India during the COVID-19 pandemic.[49] However, with the organised and freely provided government maternal healthcare, the pregnancy expenditure was expected to be at a lower level compared with the pre-COVID-19 period, since the pregnancy out-of-pocket expenditure showed a global decline of 16.2%.[50]

In this study, the pregnancy expenditure of the pregnant women delivered during the COVID-19 pandemic did not differ from the expenditure incurred for pregnant women delivered before the COVID-19 pandemic. Pregnancy expenditure is identified as an economic burden in the same study setting, since it exceeds more than 10% of household income.[25 26] This is problematic for the households whose income was affected, pushed into poverty and the pregnancy and household expenditure remained unchanged during the pandemic.

Although the Sri Lankan government provided low-income people with financial assistance (US$26.55), pregnant women's families who were pushed into poverty did not receive financial aid, and only 1.9% of families who were pushed into poverty received financial assistance. However, only 10.3% of the affected families received it, while 19.5% were already below the poverty line, adding more than 8.8% to poverty due to the pandemic. The received financial aid was a small share compared with income loss and household expenditures during the COVID-19 pandemic.

## Health services during the pandemic

During the COVID-19 pandemic, missing antenatal clinic care was reported in Saudi Arabia,[51] Sub-Saharan Africa,[52] India,[53] Pakistan[36] and almost all middle-income countries.[54] Similarly, in Sri Lanka, all routine healthcare services were interrupted due to lockdown, police curfew and travel restrictions.[21] Our study shows that the pandemic has resulted in missed antenatal clinics, yet the pregnant women were highly satisfied with the service provided by the field healthcare staff. In addition, support received from the spouses/family and neighbours was reported as 'satisfactory'. These results differ from other studies where reduced perceived family/social support was reported in the USA, Ireland and the UK.[11]

Sri Lanka's efficient public health system delivers maternal care through a well-established network of field staff and primary healthcare officers. The system provides domiciliary care and clinic-based service through grass-root level health officers, mainly via PHMs.[29] During the COVID-19 pandemic, health authorities in Sri Lanka provided interim guidelines highlighting that the PHM needs to do domiciliary visits to the infection control procedures and contact over the phone.[22 23] Our study shows that implementing those guidelines is up to the level of satisfaction of the recipients during the very early part of the pandemic (April–June 2020). The identified 'no association' between the impact of COVID-19 on the household economy and health service utilisation could also be partly attributed to the free public health system. Also, the available evidence of the region reported that families of pregnant women compromise other expenditures regarding pregnancy-related matters, and even low-income families save money for pregnancy needs.[55]

## Conclusions

During the early stage of the COVID-19 pandemic in rural Sri Lanka, the healthcare utilisation of pregnant women was minimally affected, despite the adverse impact on the household economy and healthcare delivery. However, the economic consequences pushed families into poverty, reducing nutritional expenditure. Financial aid was insufficient and not received by all in need. The postpandemic economic crisis may pose a significant threat to the Sri Lankan maternal health system, requiring targeted and strategic government and international aid. This paper highlights only the immediate impact and calls for further studies to assess the long-term effects of the pandemic.

**Acknowledgements** We acknowledge Dr Imasha Jayasinghe, Department of Community Medicine, Faculty of Medicine and Allied Sciences, Rajarata University of Sri Lanka, for the extensive support during the data collection.

**Contributors** SPG, NDW and SA were responsible for the study concept and design. SPG and TA were involved in data and project management. SPG performed data cleaning and analysis, drafted and interpreted the data and drafted the manuscript. NDW, IRP, TA and SA reviewed the manuscript. SPG is responsible for the overall content as the guarantor. All authors approved the final manuscript as submitted and agree to be accountable for all aspects of the work.

**Funding** The Accelerating Higher Education Expansion and Development (AHEAD) Operation of the Ministry of Higher Education, Sri Lanka, which was funded by the World Bank, supported this research. The funding agency has no role in the study's design, collection, analysis, interpretation of data and manuscript writing. The grant number is DOR STEM HEMS (6026-LK/8743-LK).

**Competing interests** None declared.

**Patient and public involvement** Patients and/or the public were not involved in the design, or conduct, or reporting, or dissemination plans of this research.

**Patient consent for publication** Consent obtained directly from patient(s)

**Ethics approval** This study was conducted under the large cohort study in Anuradhapura District, Sri Lanka, The Rajarata Pregnancy Cohort (RaPCo). Ethical clearance for the RaPCo study was obtained from the 'Ethics Review Committee of the Faculty of Medicine and Allied Sciences, Rajarata University of Sri Lanka (Reference no: ERC/2019/07)'. Under the Declaration of Helsinki, all participants were informed that this study was voluntary and that data handling would be confidential. All participants gave their informed verbal consent.

**Provenance and peer review** Not commissioned; externally peer reviewed.

**Data availability statement** Data are available upon reasonable request.

**ORCID iD**
Sajan Praveena Gunarathna http://orcid.org/0000-0003-0721-598X

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
