## [Reviewer comments · BMJ Open]

ARTICLE DETAILS

TITLE (PROVISIONAL)	Impact of COVID-19 Pandemic on Health Service Utilization and Household Economy of Pregnant and Postpartum Women: A Cross-Sectional Study from Rural Sri Lanka
AUTHORS	Gunarathna, Sajan; Wickramasinghe, Nuwan Darshana; Agampodi, Thilini; Prasanna, Indika Ruwan; Agampodi, Suneth

VERSION 1 – REVIEW

REVIEWER	Samuel, Laura Johns Hopkins University
REVIEW RETURNED	05-Jan-2023

GENERAL COMMENTS	The purpose of the paper is to examine self-reported household economic changes and health care utilization patterns among pregnant and post-partum women in Sri Lanka during the COVID-19 pandemic. This is an important topic. However, the paper presents a lot of analyses and lacks clarity. I suggest that the authors revise to have more hypothesis-driven analyses. Also, it seems like it would be interesting to know if the women who experienced pandemic-related economic loss were more likely to have adverse outcomes (nutrition and prenatal care utilization), but despite having the data the authors don't actually make this connection. Additional comments are provided below. The paper should be proofread throughout for grammatical errors. The abstract provides summary statistics for the sample, but the results section compares women who delivered during vs. before the pandemic. Please revise either the abstract or results section to ensure that the key results are presented consistently in the abstract and paper. The paper compares women who delivered before the pandemic to those who delivered during the pandemic. However, this comparison wasn't clear to me based on the introduction. Please provide clearer hypotheses/purpose. It's difficult to understand how to generalize these results, especially since the study used convenience sampling methods during a pandemic. Can the authors compare the sample with the target population and provide the response rate for the survey? Please add a measures section to the methods. It's not clear why measures were selected or how certain variables were measured, like financial aid (which types of aid, and duration of the recall period (year, month, etc)). Another example is nutrition expenditure – is this a measure of dietary intake? It's not clear why costs for care would change – aren't costs determined by the health care provider (or clinic, etc), not the patient? It seems that you're really interested in missed visits, which you measure separately. I think Table 3 and the accompanying text could be cut completely.
--

	I don't see nutritional expenditure results in the table – seems like it should be included in one of the tables to help the skimming reader identify this key finding. There appear to be some measures that aren't very informative. For example, there is more than one measure of poverty and the "National poverty" and "lower-middle income country" poverty limits appear to be very similar. Consider cutting one of them. Cell sizes less than 5 or maybe 10 should be avoided as a rule of thumb because small cell sizes are not meaningfully interpretable. Please collapse categories or eliminate variables altogether as appropriate. There are redundancies in the statistics which clutter the table. For example, you don't need both the test statistic (z statistic, Mann Whitney U) and the p values – just p values is sufficient. Also don't need both means and medians – just one of them. There are a lot of results here and I find it distracting. Please revise to have a more specific hypothesis-driven presentation of the results. This sentence is speculative because it extends beyond the data "Further, the services provided by PHM were highly appreciated by pregnant and postpartum women" A key limitation not acknowledged is that the selection bias may be differential based on whether the woman delivered during or before the pandemic. Since the analyses compare these two groups, it's important that the two groups be comparable, but it also seems likely that the two groups may be qualitatively or quantitatively different due to non-response during the pandemic. Do the authors have data about birth outcomes (birthweight, infant mortality, preterm birth)? It would be important to link these exposures to adverse birth outcomes.
--	---

REVIEWER	Ansari, Mohammed University of Nottingham - Malaysia Campus, School of Pharmacy
REVIEW RETURNED	08-Feb-2023

GENERAL COMMENTS	The authors have compiled a good research paper. The paper is well written with very minor grammatical mistakes. If the author can run a proof reading than the paper would be improved further. I have pointed few of those. Please quote the ethical clearance reference number and the name of the approval committee. The title suggest that many mothers were pushed into poverty due to pandemic, however after reading the manuscript it appears that percentage of people pushed into poverty were not many but those who were pushed were ably supported by the government schemes. May be the author can think of a more appropriate title. The author says that some 6% of families were pushed to poor or extreme poor condition due to covid. If the author can compare with normal data like what is percentage of people classified as poor or extremely poor under normal conditions every year as that will reflect the real impact of covid19 pandemic. The authors have listed inclusion criteria. was there any exclusion criteria also.
---

	What type of questions were asked. the authors should list the questions. How the questions were validated. Why the authors did not run an online questionnaire as it was mostly preferred during the pandemic along with an interview as some mothers may opt out of the study as they may feel uncomfortable during an interview. Did the mothers opted out of the study because of an interview if yes, would those affect the outcome of the study. The authors says that 5 mothers have to change the mode of delivery. What was the reason ..and which mode like normal to C-section or vice versa. Rewrite this line as: "pregnant women’s families who were pushed into poverty did not not receive any financial assistance..." Rewrite this as "Also, financial aid received a small share compared" as "Also, the financial aid received was a small share as compared" Please rephrase this " However, compared to the other countries, the support received from spouses/family and neighbors and the antenatal and postnatal healthcare were at a satisfactory level since reporting poor access to antenatal care and reduced perceived family/social support in the United States of America, Ireland and United Kingdom"
--	--

VERSION 1 – AUTHOR RESPONSE

Response to the Comments of Reviewer 1:Dr. Laura Samuel, Johns Hopkins University

Thank you very much for your valuable comments and suggestions on the manuscript. We have responded to each of your comments below.

Comment 1: The purpose of the paper is to examine self-reported household economic changes and health care utilization patterns among pregnant and post-partum women in Sri Lanka during the COVID19 pandemic. This is an important topic. However, the paper presents a lot of analyses and lacks clarity. I suggest that the authors revise to have more hypothesis-driven analyses.

Response: Thank you. We have removed the extended analysis done on pregnancy expenditure. The study now focuses only on a hypothesis-driven analysis to find whether the COVID-19 outbreak affects the household economy and health service utilization of pregnant and postpartum women.

Comment 2: Also, it seems like it would be interesting to know if the women who experienced pandemic-related economic loss were more likely to have adverse outcomes (nutrition and prenatal care utilization), but despite having the data the authors don’t actually make this connection. Additional comments are provided below.

Response: Thank you for this suggestion. Indeed such analysis would be valuable. We have checked the association between the health service utilization of pregnant women and the income changes during the COVID-19 pandemic as indicated below.

“There is no association between the health service utilization of pregnant women and the income changes during the COVID-19 pandemic [$\chi^2(df=1)=0.361, p=0.548$]” [Line 360-361].

Also, we checked whether there was a statistically significant difference in nutritional expenditure between families which faced income loss and their counterparts.

“There was not a statistically significant difference ($U=293.000$, $p=0.582$) of pregnancy-related nutritional expenditure during COVID-19 outbreak between families which faced income loss and their counterparts” [Line 338-341].

Comment 3: The paper should be proofread throughout for grammatical errors.

Response: Thank you. The revised manuscript was proofread before submission.

Comment 4: The abstract provides summary statistics for the sample, but the results section compares women who delivered during vs. before the pandemic. Please revise either the abstract or results section to ensure that the key results are presented consistently in the abstract and paper.

Response: Thank you very much. We completely agree with the comment since we had just presented only the data of two groups, and testing the statistical significance difference/associations are not providing any additional meaningful information. The figures of women delivered during and before the pandemic were removed from Table 1 (Socio-demographic and economic characteristics of the study sample), Table 2 (COVID-19 impact on household income and status of receiving financial aids) and from Table 4 (Impact of COVID-19 on health service utilization of pregnant and postpartum women and support received by PHM and family). However, we included the removed data as supplementary tables (supplementary tables 1, 2 and 3, respectively) for those who are interested in data/information of the two groups separately.

Comment 5: The paper compares women who delivered before the pandemic to those who delivered during the pandemic. However, this comparison wasn't clear to me based on the introduction. Please provide clearer hypotheses/purpose.

Response: Thank you. We had just presented the data for two groups in Table 1, Table 2 and Table 4. However, along with comment 4, we removed them from the tables and presented only the information related to the total sample.

The extended comparison of pregnancy expenditure breakdown between women delivered during and before the pandemic (Table 3: Third trimester per visit pregnancy expenditure between women delivered during and before the COVID-19 pandemic) was removed along with Comment 8 below. Even though it is not our primary focus, we tested whether the third trimester per visit pregnancy expenditure of women delivered during the pandemic differs from the women delivered before the pandemic (Table 2). This was done based on the similarity of socio-demographic profiles within two groups (Supplementary Table 1), and all the pregnant women who delivered during COVID-19 were in their third trimester. This information has now been updated in the manuscript [Line 258-262].

Comment 6: It's difficult to understand how to generalize these results, especially since the study used convenience sampling methods during a pandemic. Can the authors compare the sample with the target population and provide the response rate for the survey?

Response: Thank you. The Rajarata Pregnancy Cohort (RaPCo) is the largest maternal cohort to date in Sri Lanka which recruited and followed a representative sample to the rural Sri Lankan setting. We agree with the comment, this is a limitation considering a cohort study. As stated in the methods, we invited all pregnant women of the Rajarata Pregnancy Cohort (RaPCo) ($n=3,367$), but only 1,460 women responded to the telephone calls (response rate = 43.4%). Due to the unavailability of other population data, we tested whether the selected characteristics of the women of the RaPCo who did not participate in this study ($n=1,907$) differed from the present study sample ($n=1,460$) using the independent sample t-test and Chi-square goodness-of-fit test. The analysis showed that there are no statistically significant differences in relation to household income ($p=0.743$), household expenditure ($p=0.512$), religion ($p=0.363$), ethnicity ($p=0.515$), education level ($p=0.361$), and the status of sexual and reproductive health education ($p=0.344$).

We understand that this is not giving complete generalizability. Nonetheless, this study provides details of a relatively larger sample than other studies conducted during the COVID-19 pandemic. Also, we conducted two rounds and tried several attempts per round to contact and participate more women in the study.

The response rate and generalizability information was included during this revision [Line 293-299].

Comment 7: Please add a measures section to the methods. It's not clear why measures were selected or how certain variables were measured, like financial aid (which types of aid, and duration of the recall period (year, month, etc)). Another example is nutrition expenditure – is this a measure of dietary intake?

Response: Thank you very much. A detailed measures section with all the above details was added to the methods [Line 237-267].

For the status of receiving financial aid, we considered only the financial assistance (USD 26.55) provided by the Sri Lankan government. It was the only financial aid provided by the government, provided monthly for three months and only for selected families living under the national poverty line. The nutritional expenditure was the amount of money spent on pregnant women's dietary intake per month, as reported by women.

Comment 8: It's not clear why costs for care would change – aren't costs determined by the health care provider (or clinic, etc), not the patient? It seems that you're really interested in missed visits, which you measure separately. I think Table 3 and the accompanying text could be cut completely.

Response: Thank you for the comment. Pregnant women in Sri Lanka primarily use government healthcare services which provide health services free of charge at the point of service delivery. But a reasonable proportion of women use private consultations or investigations in addition to government healthcare services.

Since this pregnancy expenditure during a pandemic can cause an additional burden for households which live with pregnant women, we tested whether the pregnancy expenditure during the pandemic differs from before the pandemic. We found that per-visit pregnancy expenditure does not show a statistical significance difference. Therefore, along with this comment, we removed the extended analysis of pregnancy expenditure and added only the final cost (per visit pregnancy expenditure) without the breakdown of the expenses to Table 2.

Comment 9: I don't see nutritional expenditure results in the table – seems like it should be included in one of the tables to help the skimming reader identify this key finding.

Response: Thank you. The nutritional expenditures were added to Table 2.

Comment 10: There appear to be some measures that aren't very informative. For example, there is more than one measure of poverty and the "National poverty" and "lower-middle income country" poverty limits appear to be very similar. Consider cutting one of them.

Response: Thank you. The poverty line for lower middle-income countries" was removed and reanalyzed with the national and extreme poverty line.

Comment 11: Cell sizes less than 5 or maybe 10 should be avoided as a rule of thumb because small cell sizes are not meaningfully interpretable. Please collapse categories or eliminate variables altogether as appropriate.

Response: Thank you. We have addressed this issue by removing two columns (pregnant and postpartum women's classification), along with the previous comments, from each table.

Comment 12: There are redundancies in the statistics which clutter the table. For example, you don't need both the test statistic (z statistic, Mann Whitney U) and the p values – just p values is sufficient. Also don't need both means and medians – just one of them.

Response: Thank you. The tables were edited.

Comment 13: There are a lot of results here and I find it distracting. Please revise to have a more specific hypothesis-driven presentation of the results.

Response: Thank you. According to the previous comments, we have removed the pregnancy expenditure section. The present study focused only on the impact of the COVID-19 outbreak on the household economy and health service utilization of pregnant women.

Comment 14: This sentence is speculative because it extends beyond the data "Further, the services provided by PHM were highly appreciated by pregnant and postpartum women"

Response: Thank you. Using the Five Point Likert scale, we asked the satisfaction-related questions, where 1 denotes strongly unsatisfied, and five indicates strongly satisfied.

The above sentence was revised as follows

The majority (n=1,096, 83.3%) of the pregnant and postpartum women were satisfied with the service provided by the PHM during the pandemic [Line 351-352].

Comment 15: A key limitation not acknowledged is that the selection bias may be differential based on whether the woman delivered during or before the pandemic. Since the analyses compare these two groups, it's important that the two groups be comparable, but it also seems likely that the two groups may be qualitatively or quantitatively different due to non-response during the pandemic.

Response: Thank you. The „Strengths and limitations“ section was updated by adding the following bullet point.

□ Although we identified that the socio-demographic characteristics are not statistically different between pregnant and postpartum women in the present sample, the two groups may be qualitatively different due to non-response during the pandemic.

Comment 16: Do the authors have data about birth outcomes (birthweight, infant mortality, preterm birth)? It would be important to link these exposures to adverse birth outcomes.

Response: Thank you. Outcome data were collected separately using different sources and not through telephone interviews. We are planning a separate study on birth outcomes, and therefore we believe that it may be a repetition if we added that part to this study.

Response to the Comments of Reviewer 2: Dr Mohammed Ansari, University of Nottingham - Malaysia Campus

Thank you very much for your valuable comments and suggestions on the manuscript. We have responded to each of your comments below.

Comment 1: The authors have compiled a good research paper. The paper is well written with very minor grammatical mistakes. If the author can run a proof reading than the paper would be improved further. I have pointed few of those.

Response: Thank you very much.

Comment 2: Please quote the ethical clearance reference number and the name of the approval committee.

Response: Thank you and updated.

Comment 3: The title suggest that many mothers were pushed into poverty due to pandemic, however after reading the manuscript it appears that percentage of people pushed into poverty were not many but those who were pushed were ably supported by the government schemes. May be the author can think of a more appropriate title.

Response: Thank you. The title was changed to „Impact of COVID-19 Pandemic on Pregnant and Postpartum Women's Health Service Utilization and Household Economy: A Cross-Sectional Study from Rural Sri Lanka“.

Comment 4: The author says that some 6% of families were pushed to poor or extreme poor condition due to covid. If the author can compare with normal data like what is percentage of people classified as poor or extremely poor under normal conditions every year as that will reflect the real impact of covid19 pandemic.

Response: Thank you. The previous analysis related to poverty was updated according to the comments of reviewer one. Now 8.8% of families have been impoverished during the COVID-19 pandemic. Before the pandemic, 19.5% were below the poverty line. More 8.8% were pushed below the poverty line during the COVID-19 period. Therefore, this was updated in the discussion as follows. “Totally, 103 families (8.8%) were pushed below the poverty line, while 19.5% of families already remained in the poorest group” [Line 376-377].

Comment 5: The authors have listed inclusion criteria. Was there any exclusion criteria also?

Response: Thank you. We invited all eligible pregnant women already recruited for the RaPCo study. The exclusion criteria are included below from the RaPCo study.

“Pregnant women with uncertain due dates and those who planned to leave the study setting for delivery were excluded from the study” [Line 209-210].

Comment 6: What type of questions was asked? The authors should list the questions.

Response: Thank you. We asked the questions based on five categories in the Data Analysis section.

Also, we have added the interview guide as a supplementary file (Supplementary File 1).
 “Data collection was based on five major categories: 1) whether income was affected during the outbreak, 2) financial assistance received, 3) the status of utilizing maternal health services, 4) the assistance of the PHM, and 5) the assistance of household members/neighbors (Supplementary File 1)” [Line 221-225].

Comment 7: How the questions were validated. Why the authors did not run an online questionnaire as it was mostly preferred during the pandemic along with an interview as some mothers may opt out of the study as they may feel uncomfortable during an interview. Did the mothers opted out of the study because of an interview if yes, would those affect the outcome of the study.

Response: Thank you. We pre-tested and edited the interviewer guide according to the pre-tested sample response.

Although an online questionnaire is the most preferred way during a pandemic, it is not feasible in a lowresource rural setting, mainly because only 476 (32.6%) women in the sample use internet and messaging apps. We believe that if we use an online questionnaire in this setting, the response rate might be much lower than the current rate.

During the telephone survey, none of the participants opt-out. We believe this was because female medical graduates trained in telephone interview techniques, emphasizing politeness, telephone etiquette, basic counselling skills, and essential perinatal health information, carried out the interviews. Also, during the telephone interviews, we provided psychological support to pregnant and postpartum women where needed. In addition, the cohort was followed up by the research team from early pregnancy. We assume that opting out would be rare as the women had the opportunity to build trust with the research team and the institution.

Comment 8: The authors say that 5 mothers have to change the mode of delivery. What was the reason and which mode like normal to C-section or vice versa.

Response: Thank you. The change from normal vaginal delivery to lower segment C-sections and pregnant women claimed that the change had to be done due to the COVID-19 pandemic. However, we did collect data regarding the documented evidence for the actual reason.

Comment 9: Rewrite this line as: "pregnant women"s families who were pushed into poverty did not receive any financial assistance..."

Rewrite this as "Also, financial aid received a small share compared" as "Also, the financial aid received was a small share as compared"

Response: Thank you. These sentences were updated as suggested.

Comment 10: Please rephrase this "However, compared to the other countries, the support received from spouses/family and neighbours and the antenatal and postnatal healthcare were at a satisfactory level since reporting poor access to antenatal care and reduced perceived family/social support in the United States of America, Ireland and United Kingdom"

Response: Thank you and rephrased as follows.

“the support received from spouses/family and neighbours was reported as satisfactory. These results are somewhat different from other studies where reduced perceived family/social support was reported in the United States of America, Ireland and the United Kingdom.”

VERSION 2 – REVIEW

REVIEWER	Samuel, Laura Johns Hopkins University
REVIEW RETURNED	02-May-2023
GENERAL COMMENTS	The authors have addressed my concerns. I have no further comments.